Chaetoglobosin A induces apoptosis in T-24 human bladder cancer cells through oxidative stress and MAPK/PI3K-AKT-mTOR pathway

Song Jia 1
Qiao Jinyu 1
Chen Mingxue 1
Li Jiahui 1
Wang Jixia 2
Yu Dayong 3
Zheng Huachuan 4
Shi Liying 1 shiliying@dlu.edu.cn
1 School of Life and Health, Dalian University , Dalian , China
2 Dalian Institute of Chemical Physics, Chinese Academy of Sciences , Dalian , China
3 School of Basic Medical Sciences, Chengde Medical University , Chengde , China
4 Department of Oncology, The Affiliated Hospital of Chengde Medical University , Chengde , China
Sarkar Ripon
Electronic publication date: 2025 Mar 31
Publication date: 2025
Volume: 13
Electronic Location ID: e19085
Received 2024 Apr 17; Accepted 2025 Feb 11
Copyright: © 2025 Song et al.
Copyright year: 2025
Copyright holder: Song et al.
License: This is an open access article distributed under the terms of the Creative Commons Attribution License, which permits unrestricted use, distribution, reproduction and adaptation in any medium and for any purpose provided that it is properly attributed. For attribution, the original author(s), title, publication source (PeerJ) and either DOI or URL of the article must be cited.
License URL: https://creativecommons.org/licenses/by/4.0/

Keywords: Chaetoglobosin A, Apoptosis, MAPK pathway, PI3K-AKT-mTOR pathway, Reactive oxygen species

Funding: Natural Science Basic Research Project of Education Department of Liaoning Province LJKMZ20221837 CAS Key Laboratory of Separation Sciences for Analytical Chemistry Dalian Institute of Chemical Physics Chinese Academy of Sciences KL2208 The Technical Research Plan of Liaoning Province Science and Technology Joint Program 2024JH2-102600065 This research was funded by the Natural Science Basic Research Project of Education Department of Liaoning Province (LJKMZ20221837) and CAS Key Laboratory of Separation Sciences for Analytical Chemistry, Dalian Institute of Chemical Physics, Chinese Academy of Sciences (No. KL2208). The Technical Research Plan of Liaoning Province Science and Technology Joint Program (2024JH2-102600065) supported the APC. The funders had no role in study design, data collection and analysis, decision to publish, or preparation of the manuscript.

==============================
Chaetoglobosin A (ChA) is an antitumor compound produced by Chaetomium globosum. However, the mechanism of its antitumor effect has been rarely reported. In this study, we evaluated the anti-proliferative effect of ChA on T-24 human bladder cancer cells and explored its mechanism of action. ChA was found to have a good inhibitory effect on T-24 cells by MTT assay with an IC50 value of 48.14 ± 10.25 μΜ. Moreover, it was found to have a migration inhibitory ability and a sustained proliferation inhibitory effect on tumor cells by cell aggregation assay and cell migration assay. The cells morphological changes were determined by Hoechst33342 assay. While Annexin V-FITC/PI double-staining assay also demonstrated that the number of apoptotic cells increased with the increase of drug concentration. Flow cytometry results showed that ChA treatment increased reactive oxygen species (ROS) and decreased mitochondrial membrane potential (MMP) in T-24 cells and inhibited cell mitosis, resulting in an increase in the number of sub-G1 phase cells. Further western blot experiments demonstrated that MAPK and PI3K-AKT-mTOR pathways were activated after drug treatment in addition to endogenous and exogenous apoptotic pathways. The addition of the ROS inhibitor N-acetylcysteine (NAC) upregulated the expression level of Bcl-2 protein, decreased p38 phosphorylation, increased ERK phosphorylation and restored the levels of PI3K and p-mTOR after ChA treatment. These suggest that ChA induces apoptosis by regulating oxidative stress, MAPK, and PI3K-AKT-mTOR signaling pathways in T-24 cells.

Introduction

Bladder cancer is one of the top 10 most common types of cancer worldwide. Each year, approximately 600,000 people worldwide are diagnosed with bladder cancer and more than 200,000 die from the disease (Sung et al., 2021). Diagnosis of bladder cancer relies heavily on cystoscopy which is an expensive procedure. Also, treatment outcomes are poor, therefore, bladder cancer is the most challenging and expensive to diagnose and treat (Svatek et al., 2014). Transurethral resection of bladder tumors (TURBT) is the mainstay of treatment for patients with bladder cancer, and despite some surgical and anesthetic improvements, long-term patient survival rates have remained constant for decades (Brausi et al., 2002). Therefore, the development of a new anti-bladder cancer treatment remains a great challenge for researchers and healthcare providers. There is a belief that treatment will be personalized, at the same time, our understanding of the disease has increased dramatically through advanced molecular biology research. Indeed, cisplatin-based combination neoadjuvant chemotherapy has been shown to prolong overall survival in patients undergoing cystectomy and has become routine care (Vale, 2005). N-acetylcysteine (NAC) has provided survival benefits for patients with clinical T2, T3 and even T4 bladder cancer. Tumor cell-associated survival pathways were shown to be associated with survival in bladder cancer (Grossman et al., 2003). ADAMTS9-AS1 promotes the invasion and migration of bladder cancer cells and negatively regulates the apoptosis and autophagy of bladder cancer cells, possibly through the PI3K/AKT/mTOR signaling pathway (Yang et al., 2021). Lemur tyrosine kinase 3 (LMTK3) as a bladder cancer oncogene has been shown to promote bladder cancer cell proliferation and migration through ERK/MAPK pathway (Jiang et al., 2020).

Microbial metabolites are an important source for the discovery of drug lead compounds. Chaetoglobosins are fungal secondary metabolites belonging to cytochalasin alkaloids, and researchers have extracted trichothecene from the secondary metabolites of fungi and found to have biological activities including antitumor (Carlsson & Simonsen, 2015), antifungal (Johansen & Lamark, 2020), antibacterial (Mancias & Kimmelman, 2016), anti-inflammatory (Stolz, Ernst & Dikic, 2014), fibrinolytic activity (Tanida et al., 2005), anti-HIV activity (Huang et al., 2016) and other biological activities. Therefore, they have promising applications that have attracted researchers to further study them. ChA one of the cytochalasin analogues, has been found to be significantly cytotoxic to the following cell lines with 72 h drug treatment: A549 cell line IC50 = 6.56 mol/L, SGC-7901 cell line IC50 = 7.48 mol/L, MDA-MB-435 cell line IC50 = 37.56 mol/L, HepG2 cell line IC50 = 38.62 mol/L, HCT116 cell line IC50 = 3.15 mol/L, and P388 cell line IC50 = 3.67 mol/L (Huang et al., 2016; Nath et al., 2014; Jiang et al., 2017). Regarding the antitumor mechanism, it was found that ChA preferentially induces apoptosis in chronic lymphocytic leukemia (CLL) cells under culture conditions that mimic lymphoid tissue. The mechanism of action study revealed that ChA targets filamentous actin in CLL cells, thereby inducing cell cycle arrest and inhibiting cell migration, indicating that this compound is a potential CLL drug (Knudsen et al., 2014). The above studies suggest that ChA can lead to the death of some tumor cell lines. Since the anti-tumor mechanism of action of ChA is less reported, it is necessary to encourage researchers to investigate it further.

This study investigated the antitumor effect of ChA on bladder cancer T-24 cells and its molecular mechanism. It was found that ChA inhibited T-24 cell proliferation and caused cell cycle arrest, and activated MAPK pathway as well as PI3K-AKT-mTOR pathway to induce apoptosis.

Materials and Methods

Reagents and antibodies

Trypsin, MTT, and dimethyl sulfoxide (DMSO) were purchased from Sangon Biotech (Shanghai, China). Annexin-V-fluorescein isothiocyanate (FITC), RNase, propidium iodide (PI), 2′,7′-Dichlorodihydrofluorescein diacetate (DCFH-DA), 2′-(4-ethoxyphenyl)-5-(4-methyl-1-piperazinyl)-2,5′-bi-1H-benzimidazole, trihydrochloride (Hoechst 33342), 3,3′-dihexyloxacarbocyanine iodide (DioC6) and NAC were purchased from Sigma Aldrich (St. Louis, MO, USA). Chloroquine (CQ), SB203580, 3-methyladenine (3-MA) and rapamycin (Rap) purchased from Aladdin. Portions of this text were previously published as part of a preprint (https://doi.org/10.21203/rs.3.rs-3258565/v1).

Cell culture

Human bladder cancer cell line T-24 was provided by the cell bank of the Chinese Academy of Sciences in Shanghai. Cells were cultured in DMEM medium (Gibco, Waltham, MA, USA) enriched with 10% FBS (Gibco, Waltham, MA, USA) at 37 °C with 5% CO2 and passaged at 80–90% confluence. T-24 cell in the logarithmic growth phase was used for the following experiments.

Measurement of cytotoxicity and colony formation assay

Cells (5,000/well) were inoculated into 96-well plates in 100 μL of culture medium. Cytotoxicity was examined using the MTT assay. The cells were treated with ChA at different concentrations (0, 12.5, 25, 50, 75, 100 μM) for 24 h. After the desired treatment, cells were incubated with 10 μL of MTT solution for an additional 4 h. Add 150 μL DMSO to dissolve formazan and incubate at 37 °C overnight. Cell absorbance at 570 nm was measured using the Universal Microplate Spectrophotometer (Bio-Rad, Berkeley, CA, USA).

Cells (600/well) were inoculated into 6-well plates and cultured in fresh medium containing ChA or DMSO, with media changes every 3 days. After 14 days, cells were fixed for 30 min in PFA 4% and crystal violet staining was performed at room temperature (RT). Repeatedly rinsed and photographed under the microscope.

Wound healing assay

For easy observation, we drew horizontal lines at 1 cm intervals on the back of the six-well plate. Cell plate count method was used to adjust the cell concentration to about 5 × 105/mL and the cells were inoculated into a six-well plate. When the cell density reached approximately 80%, the monolayer was wounded by scratching with a 10 µl sterile pipette tip lengthwise along the plate surface, and the media was removed. The cells were then washed three times with PBS and cultured in serum-free media. The cells were treated with ChA at different concentrations (0, 1.56, 3.125, 6.25, 12.5 μM) for 24 h. Imaging was done at 0 and 24 h under light microscopy.

Hoechst 33342 analysis

After ChA treatment in different concentrations for 6 h, cells were collected and washed with PBS, fixed with 4% paraformaldehyde for 30 min at 4 C, and stained with 1 mM Hoechst 33258 for 5 min. Nuclei and apoptotic bodies were observed under a fluorescence microscope (Nikon, Tokyo, Japan) with ×20 objective lens and ×10 eyepiece and photographed.

Apoptosis assay

T-24 cells were firstly stimulated with an increasing concentration of ChA solution (0, 25, 50, 75 µM). After 12 h treatment, the cells were collected by centrifugation after trypsinization, and washed with cold PBS. Cultured cells were resuspended and stained with Annexin V-FITC and PI in binding buffer according to the kit’s protocol and the cells were examined by flow cytometry (BD Biosciences, Franklin Lakes, NJ, USA) within one hour.

Measurement of ROS

T-24 cells were firstly stimulated with an increasing concentration of ChA solution (0, 25, 50, 75 µM) for 2 h. After stimulation, T-24 cells were digested by trypsin and collected by centrifugation. Then the cells were stained with 5 μM DCFH-DA for 30 min. After two washes with PBS to remove the excess DCFH-DA, the population of stained cells was examined by flow cytometry (BD Biosciences, Franklin Lakes, NJ, USA).

Measurement of MMP

T-24 cells were firstly stimulated with an increasing concentration of ChA solution (0, 25, 50, 75 µM) for 4 h. After stimulation, T-24 cells were digested by trypsin and collected by centrifugation. Then the cells were washed in PBS and treated with 40 nM DioC6(3) for 30 min. Subsequently, the percentage of cells with MMP loss was examined by flow cytometry (BD Biosciences, Franklin Lakes, NJ, USA).

Cell cycle analysis

T-24 cells were firstly stimulated with an increasing concentration of ChA solution (0, 25, 50, 75 µM). After 6 h treatment, the cells were collected by centrifugation after trypsinization, and washed with cold PBS. Cells were resuspended in 1 mL PBS containing 20 μg/ml RNase A, incubated at 4 C for 30 min, and resuspended in propidium iodide (PI) solution. Cells at different stages of the cell cycle and sub-G1 phase were analyzed by flow cytometry (BD Biosciences, Franklin Lakes, NJ, USA).

Western blot analysis

T-24 cells were firstly stimulated with an increasing concentration of ChA solution (0, 25, 50, 75 µM). After 6 h treatment, cells were washed with PBS three times, and the total protein was extracted using RIPA lysis buffer or RIPA lysis buffer containing protease and phosphatase inhibitors. After brief sonication, the lysates were centrifuged at 13,000 × g for 10 min at 4 °C, and the protein contents in the supernatant were measured by a BCA kit (Solarbio, Beijing, China). Proteins were separated on dodecyl sulfate, sodium salt-polyacrylamide gel electrophoresis (SDS-PAGE), transferred on poly-vinylidene fluoride (PVDF) membranes, and blocked with 5% skimmed milk for 1 h. The membranes were then probed with primary antibodies overnight at 4 °C, followed by secondary antibody for 2 h at room temperature. For the detection of proteins, the chemiluminescence agents ECL kit (Beyotime, Shanghai, China) was used, and the protein expression was analyzed by Image J software. Western blot analysis was performed using the following specific monoclonal or polyclonal antibodies: anti-Bcl-2 (Cat# sc-73822), anti-Bax (Cat# sc-20067), anti-Bid (Cat# sc-56025), anti-CDK1 (Cat# sc-53219), anti-CCNB1 (Cat# sc-53238), anti-ERK (Cat# sc-81459), anti-JNK (Cat# sc-7345) and anti-GAPDH (Cat# sc-137179) from Santa Cruz, Dallas, Texas, USA; caspase-8 (Cat# AF6442), anti-phospho-ERK1/2 (Cat# AF3240), anti-phospho-JNK (Cat# AF3319), anti-PI3K (Cat# AF6241), anti-p-PI3K p85 alpha (Tyr607) (Cat# AF3241), anti-AKT (Cat# AF0836), secondary anti-mouse (Cat# SA00001-1) and anti-rabbit antibodies (Cat# SA00001-2) from Affinity, Cincinnati, OH, USA; anti-p-AKT (Ser473) (Cat#66444-1-1g), anti-mTOR (Cat#66888-1-1g), anti-p-mTOR (Ser2448) (Cat#67778-1-1g), anti-Calnexin (Cat#10427-2-AP) from Proteintech Group, Chicago USA; anti-p38 (Cat#9212S) and anti-phospho-p38 (Cat#9211S) from Cell Signaling Technology (Danvers, MA, USA).

Statistical analysis

All experiments were repeated at least three times independently. The data were expressed as the mean ± standard deviation (SD). The treated groups were compared by one-way variance (ANOVA) with SPSS 13.0 (SPSS, Chigaco, IL, USA). The statistically significant p values were labeled as follows: *p < 0.05, **p < 0.01, ***p < 0.001.

Results

Inhibitory effect of ChA on T-24 cell proliferation

To investigate the proliferation inhibitory effect of ChA on tumor cells, we treated T-24 cells with different concentrations of ChA (0, 12.5, 25, 50, 75, 100 μM) for 24 h. Figure 1A is the structural formula of ChA. The results showed that ChA could reduce cell viability in a dose-dependent manner as measured by MTT (Fig. 1B). We analyzed the effect of ChA (0, 1.56, 3.12, 6.25, 12.5, 25 μΜ) on the cloning ability of T-24 cells at lower concentration ranges. The results showed that the clonogenic “spots” were reduced at lower concentrations compared to the unspoked group, and almost no clonogenic “spots” were formed after the drug concentration reached 12.5 μM (Fig. 1C). These results indicate that ChA can significantly inhibit the proliferation and division of T-24 cells and has a sustained inhibitory effect.

Figure 1 ChA treatment inhibited cell proliferation in T-24 cells.

(A) The chemical structures of ChA. (B) T-24 cells were pretreated with different concentrations or no concentration of ChA, and then cell viability was detected by MTT after 24 h. (C) The effect of ChA on colony formation. All data are presented as mean ± SD. ***p < 0.001.

ChA inhibits T-24 cell migration

The migration ability of the cells could be examined by the number of cells entering the blank area at the edge of the scratch. As shown in the Figs. 2A and 2B, the changes of cell migration at 24 h after different concentrations of ChA treatment. The distance of cell scratch was the same in all groups at 0 h, and after 24 h of treatment, the cells at the edge of the scratch gradually entered the blank area. Compared with the control group, as the concentration of the drug increased, the cell migration ability decreased. The above results indicated that ChA could inhibit the migration of T-24 cells.

Figure 2 ChA inhibits T-24 cell migration.

(A) Photographs of inhibition of T-24 cell migration after treatment with various concentrations of drugs. (B) Statistical results of inhibition of migration rate of T-24 cells after treatment with various concentrations of drugs. All data are presented as mean ± SD. ***p < 0.001.

ChA induces apoptosis in T-24 cells

It was found that ChA had a high cytotoxic effect on human bladder cancer T-24 cells by pre-MTT experiments. In order to explore whether ChA killed tumor cells through the apoptotic pathway, Hoechst 33342 fluorescence staining was performed on T-24 cells treated with different concentrations of ChA, and the results are shown in Fig. 3A. Compared with the unspiked group, ChA-treated T-24 cells with increasing drug concentration the fluorescence intensity increased and the apoptotic morphological features of cell crumpling and chromatin condensation appeared, indicating that ChA induced apoptosis in T-24 cells (Nitobe et al., 2003). The quantified and the flow through results are shown in Figs. 3B and 3C to determine the specific apoptotic rate. The total apoptotic rate of T-24 cells treated with different concentrations of ChA increased in a concentration-dependent manner compared to the untreated group and increased significantly at drug concentrations up to 75 μM (p < 0.001), which is consistent with the morphological observations.

Figure 3 ChA induces apoptosis in T-24 cells.

(A) After T-24 cells were treated with ChA for 6 h and stained with Hoechst 33342, morphological changes were observed by fluorescence microscopy. (B) For T-24 cells after 12 h of ChA treatment, the cells were treated with annexin V/FITC and PI for staining. The rate of apoptosis was detected by flow cytometry, and the ratio of early apoptotic cells (lower right quadrant) and late apoptotic cells (upper right quadrant) were summed. (C) Statistical results of apoptosis detection by flow cytometry. **p < 0.01, ***p < 0.001, compared with 0 μM group.

ChA induces ROS accumulation and MMP reduction in T-24 cells

The accumulation of intracellular reactive oxygen species (ROS) is associated with the development of several diseases and is also closely linked to apoptosis (Wang et al., 2023; Dröge, 2002). In this experiment, the detection of intracellular ROS was performed by fluorescent probe (DCFH-DA). Flow cytometry results showed that intracellular ROS levels were elevated after 2 h of dosing treatment and significantly higher at 75 μM (p < 0.001) compared to the non-dosed group. In addition, the addition of NAC pretreatment restored ROS levels after ChA treatment (Figs. 4A and 4B). The decrease in the MMP is an event that occurs early in apoptosis (Cui et al., 2014). We used DiOC6(3) dye for mitochondrial membrane potential detection. A decrease in green fluorescence intensity indicating MMP loss was detected by flow cytometry (Figs. 4C and 4D). The degree of loss of intracellular mitochondrial membrane potential was significantly increased after the drug concentration reached 50 μM compared to the unspiked group (p < 0.001). This indicates that excessive accumulation of ROS in T-24 cells leads to oxidative stress and causes mitochondrial dysfunction, inducing apoptosis in T-24 cells.

Figure 4 ChA treatment induces ROS production, MMP loss in T-24 cells.

(A) ROS production in T-24 cells treated with DCFH-DA for 2 h before staining with ChA. y-axis indicates the number of cells. x-axis indicates fluorescence intensity. (B) Proportion of cells with increased fluorescence intensity (%). (C) MMP deletion was detected by DioC6 (3) staining after 4 h of ChA treatment. (D) MMP deletion quantification results. *p < 0.05, **p < 0.01, ***p < 0.001, compared with 0 μM group.

ChA induces T-24 cell cycle blockade

To explore the mechanism of ChA inhibition of T-24 cell growth, the distribution of T-24 cells cycle and the relative changes of each cycle after 6 h of drug treatment were examined and analyzed separately in this study. The flow results (Figs. 5A and 5B) showed that the Sub G1 (subdiploid DNA) cell population increased after ChA treatment, with a slight increase in G1 phase, a less pronounced change in S phase, a slight decrease at 75 μM, and a gradual G2/M phase cell decreased. This indicates that ChA treatment increased DNA fragmentation, caused arrest of G1 phase cells and inhibited cell mitosis.

Figure 5 Effect of ChA on the T-24 cells cycle.

(A) Flow cytometry was performed to detect the cell cycle distribution in T-24 cells after 6 h of ChA treatment. (B) Effect of ChA treatment on the various cell cycles of T-24 cells. (C) After treatment of T-24 cells with ChA for 6 h, changes in CDK1, CCNB1 protein expression were detected by western blot. (D) Protein expression levels were quantified and normalized to GAPDH. Values are expressed as mean ± SD of three independent experiments. *p < 0.05, **p < 0.01, ***p < 0.001, compared with 0 μM group.

Changes in the expression of CCNB1 and CDK1, G2/M phase cell cycle regulatory proteins that play an important role in cell cycle progression, were analyzed by western blot. As shown in the Figs. 5C and 5D, the expression of CDK1, CCNB1 was decreased in a dose-dependent manner after ChA treatment of T-24 cells for 6h, which prevented the formation of CCNB1-CDK1 complex and further hindered the cell mitosis.

Effect of ChA on apoptosis-associated proteins in T-24 cells

The effect of ChA on MMP was detected by flow cytometry, and it was found that the dysregulation of Bcl-2 family proteins was found to directly affect MMP (Hassan et al., 2014; Nasser et al., 2017). As shown in Figs. 6A and 6B, increased Bax expression and decreased Bcl-2 expression were observed in ChA-treated T-24 cells, consistent with decreased MMP detected by flow, suggesting that the Bcl-2 family is involved in the mitochondrial apoptotic pathway. Whereas caspase family proteins are closely linked to Bcl-2 family proteins in apoptosis, we examined the initiating caspase (caspase-8) and effector caspase (caspase-3) as well as Bid proteins. The results were as follows: caspase-3 and caspase-8 protein expression showed a decreasing trend, and activated caspase-8 broke Bid in the cytoplasm, leading to Bid protein cleavage further affecting MMP. The results indicate that ChA can induce apoptosis in T-24 cells by way of mitochondrial pathway and death receptor pathway.

Figure 6 Effect of ChA on T-24 apoptosis protein.

(A) Expression of caspase-3, caspase-8, Bid, Bax and Bcl-2 were analyzed by western blot. (B) Statistical analysis. Expression of proteins level were quantified and standardized against GAPDH. Values are expressed as the mean ± SD of three independent experiments. *p < 0.05, **p < 0.01, ***p < 0.001, compared with 0 μM group.

ChA induces activation of MAPK pathway in T-24 cells

The mitogen-activated protein kinase MAPK signaling pathway plays an important role in a series of cellular physiological activities such as tumor cell growth, differentiation and apoptosis. Among them, JNK and p38 MAPK functions are associated with inflammatory factors and oxidative stress, and ERK1/2 is mainly involved in cell growth, differentiation and other activities (Wagner & Nebreda, 2009). To verify whether ChA induces apoptosis through MAPK pathway regulation, this experiment detected JNK, p38 as well as ERK1/2 and their phosphorylated protein expression levels by western blot and the results are shown in Figs. 7A and 7B. Compared with the unspiked group, the levels of JNK and p38 phosphorylation were significantly increased after 6 h of drug treatment, and ERK1/2 phosphorylation levels were significantly increased when the drug concentration reached 75 μM. This indicates that ChA induces apoptosis in T-24 cells by regulating the MAPK pathway.

Figure 7 Regulation of MAPK as well as PI3K-Akt-mTOR signaling pathway proteins in ChA-treated T-24 cells.

(A) Western blotting detection of ERK/p38/JNK and p-ERK/p-p38/p-JNK protein levels in T-24 cells. (B) Statistical analysis. Expression of proteins level were quantified and standardized against GAPDH. (C) Western blotting detection of PI3K/AKT/mTOR and p-PI3K/p-AKT/p-mTOR protein levels in T24 cells. (D) Statistical analysis. Expression of proteins level were quantified and standardized against calnexin. Values are expressed as the mean ± SD of three independent experiments. ***p < 0.001, ****p < 0.0001, compared with 0 μM group.

Effect of ChA on PI3K-Akt-mTOR pathway

The PI3K-Akt-mTOR signaling pathway is thought to play a key role in regulating proliferation, migration, and autophagy. Dysregulation of PI3K-Akt-mTOR signaling was found in 40% of bladder cancers (Khan et al., 2020). To detect the effect of PI3K-AKT-mTOR pathway changes on bladder cancer T-24 cell line after compound ChA treatment, the experiment used western blot to detect PI3K, AKT, and mTOR proteins, and selected phosphorylation sites to detect p-PI3K-p58, p-AKT-Ser473, and p-mTOR-Ser2448 expression changes. The experimental results are shown in the Figs. 7C and 7D. Compared with the unspiked group, the compound ChA spiked treatment could increase the levels of p-PI3K-p58 and p-AKT-Ser473, and inhibit the phosphorylation expression of mTOR protein. The above experimental results suggest that the PI3K-AKT-mTOR pathway is involved in regulating compound ChA-induced apoptosis in T-24 cells.

Effect of inhibitors on protein expression in T-24 cells after ChA treatment

To further investigate the key roles played by ROS, p38, mTOR, and autophagy in ChA-induced apoptosis in T-24 cells, we analyzed the effects of drug combination inhibitors on intracellular protein expression of related pathways by western blot. We used co-incubation with the corresponding inhibitors before ChA treatment of cells. The inhibitors and the concentrations used were reactive oxygen species inhibitor NAC (5 mM), mTOR inhibitor Rapamycin (5 μM), p38 MAPK inhibitor SB203580 (20 μM) and PI3K inhibitor 3-MA (5 mM) were incubated for 1 h before ChA treatment, and then cells were treated with 75 μM of ChA for 6 h.

Western blot results (Fig. 8) showed that NAC pretreatment upregulated the expression level of Bcl-2 protein and decreased p38 phosphorylation and increased ERK phosphorylation. ROS was also found to be involved in the mTOR signaling pathway, and the levels of PI3K and p-mTOR were significantly restored after NAC pretreatment with ChA. ROS was found to negatively regulate Bcl-2, MAPK, and PI3K-Akt-mTOR signaling pathways by adding the ROS inhibitor NAC.

Figure 8 Effect of inhibitors on protein expression in T-24 cells after ChA treatment. T-24 cells were pretreated with NAC (5 mM), Rapamycin (5 μM), SB203580 (20 μM) and 3-MA (5 mM) for 1 h, and then treated with 75 μM ChA for 6 h.

(A) Typical western blot bands for Bcl-2, p-p38, p38, p-ERK, ERK, p-PI3K, PI3K, p-mTOR, and mTOR. (B) Bcl-2 protein expression statistics, (C) p-ERK/ERK protein expression statistics, (D) p-p38/p38 protein expression statistics, (E) p-PI3K/PI3K protein expression statistics, (F) p-mTOR/mTOR protein expression statistics. Values are expressed as the mean ± SD of three independent experiments. Compared with Control group, ***p < 0.001, compared with ChA 75 μM group, ###p < 0.001.

Pretreatment with PI3K inhibitor 3-MA and mTOR inhibitor Rapamycin had no effect on the expression levels of Bcl-2 protein in the mitochondrial pathway and ERK in the MAPK pathway, but could up regulate the expression of PI3K. P38MAPK inhibitor SB203580 pretreatment decreased the level of p38 phosphorylation and increased the phosphorylation of ERK. Figs. 8B–F shows the statistical diagram of western blot results. Through the analysis of the above experimental results, we deduced that ChA treatment of T-24 cells resulted in the accumulation of intracellular ROS as an important factor leading to apoptosis. When inhibiting the production of ROS, it could simultaneously affect the MAPK pathway, mTOR pathway and mitochondria-regulated endogenous pathways jointly involved in the regulation of ChA-induced apoptosis in T-24 cells. Among where the interaction of the MAPK pathway and the mTOR pathway was not confirmed, but the joint participation of multiple pathways amplifies the apoptotic cascade response in T-24 cells.

Discussion

Surgery and chemotherapy were used as early means of treating tumors, but due to their high damage to the organism, new treatment modalities are urgently needed. Microbial metabolites are an important source for the discovery of drug lead compounds, and natural products have become the focus of their fight against various cancers with fewer side effects and less drug resistance than conventional chemotherapy and radiotherapy (Atanasov et al., 2021; Keller, 2019). Although ChA has been known for more than 40 years and has received considerable attention for its anticancer activity, research on its pharmacological activity is still limited (Brewer & Taylor, 1978; Ohtsubo et al., 1978).

In the present study, ChA showed a dose-dependent inhibitory effect on T-24 cells. Moreover, Colony formation assay as well as cell migration assay revealed that ChA inhibits T-24 cell viability and has migration inhibitory ability and sustained proliferation inhibition. A link has been found between the exploration of cell death mechanisms and targeted cancer therapy (Carlsson & Simonsen, 2015; Johansen & Lamark, 2020), and the effective elimination of cancer cells through apoptosis has been the main goal and trend of current clinical cancer therapy. The characteristics of apoptosis included specific morphological as well as biochemical alterations (Pistritto et al., 2016). The appearance of apoptotic morphology after ChA treatment was observed by Hoechst 33342 staining, while Annexin V-FITC/PI double staining assay also showed that the number of apoptotic cells increased with increasing drug concentration. Apoptosis occurs mainly through endogenous and exogenous pathways (Fulda et al., 2010; Carneiro & El-Deiry, 2020; Hassan et al., 2014). A dose-dependent increase in the ratio of Bax to Bcl-2 was observed in ChA-treated T-24 cells by western blot, indicating that the Bcl-2 family is involved in the mitochondrial pathway of apoptosis. While the expression of initiating caspase (caspase-8) and effector caspase (caspase-3) proteins showed a decreasing trend, activated caspase-8 broke Bid in the cytoplasm, leading to a decrease in Bid expression, which further affected the decrease of mitochondrial membrane potential. The results indicated that ChA could induce apoptosis in T-24 cells through both endogenous and exogenous pathways.

ROS are partially reduced metabolites of oxygen with a strong oxidative capacity that are involved in the regulation of normal cellular metabolic processes and are required for the maintenance of endostasis and cell signaling (Thannickal & Fanburg, 2000). Several studies have identified that high levels of ROS have the ability to induce cell cycle arrest, senescence, and cancer cell death by initiating apoptotic signaling within the cell cycle or through the intra-mitochondrial apoptotic signaling and death receptor pathways (Gao et al., 2020). Furthermore, when ROS inhibited the phosphatase CCNB1, leading to the activation of CDK1, it promoted the cycle progression of cancer cells (Takeuchi et al., 2005; Guertin & Sabatini, 2007). Flow cytometry results showed that ChA treatment increased reactive oxygen levels in T-24 cells, inhibited cell mitosis, and increased the number of sub-G1 phase cells. The increase in ROS can also be controlled through PI3K/AKT and MAPK pathways (Salmeen et al., 2003; Allegra et al., 2020). Major redox-active signaling pathways such as the p38MAPK pathway, a major sensor of cellular redox status, have been shown to promote apoptosis and suppress tumor genesis by inhibiting ROS production in malignant cells (Lotem et al., 1996). The p38 inactivation leads to intracellular ROS accumulation, and excessive ROS levels further cause pathway cascade effects. Enhanced phosphorylation of p38 and JNK after ChA treatment, reduced ERK phosphorylation triggered activation of MAPK pathway, and elevated levels of PI3K and AKT phosphorylation and downregulation of p-mTOR protein expression demonstrated the involvement of PI3K-AKT-mTOR pathway. Finally, combined treatment with inhibitors demonstrated that NAC significantly restored p38 phosphorylation and decreased ERK phosphorylation expression by eliminating excess ROS, while other inhibitors were not effective in the pathway. Thus, ChA-induced oxidative stress can further regulate MAPK pathways and subsequently induce apoptosis, and the PI3K-AKT-mTOR pathway is jointly involved in the regulation of T-24 cell apoptosis.

In summary, ChA treatment leads to cycle blockage inhibiting T-24 cell proliferation and generating oxidative stress activating MAPK and PI3K-AKT-mTOR pathways to induce apoptosis in T-24 cells for antitumor effects. The results provide an experimental basis for the development of novel antitumor drugs with cytochalasin-like compound Chaetoglobosin A.

Supplemental Information

Supplemental Information 1 Colony number.

Supplemental Information 2 Raw data.

Additional Information and Declarations

Competing Interests

The authors declare that they have no competing interests.

Author Contributions

Jia Song performed the experiments, prepared figures and/or tables, and approved the final draft.

Jinyu Qiao performed the experiments, prepared figures and/or tables, and approved the final draft.

Mingxue Chen performed the experiments, prepared figures and/or tables, and approved the final draft.

Jiahui Li performed the experiments, prepared figures and/or tables, and approved the final draft.

Jixia Wang analyzed the data, authored or reviewed drafts of the article, and approved the final draft.

Dayong Yu conceived and designed the experiments, prepared figures and/or tables, and approved the final draft.

Huachuan Zheng conceived and designed the experiments, analyzed the data, authored or reviewed drafts of the article, and approved the final draft.

Liying Shi conceived and designed the experiments, analyzed the data, authored or reviewed drafts of the article, and approved the final draft.

Data Availability

The following information was supplied regarding data availability:

The raw measurements are available in the Supplemental File.

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
