# Peer review of "Chaetoglobosin A induces apoptosis in T-24 human bladder cancer cells through oxidative stress and MAPK/PI3K-AKT-mTOR pathway"

_PeerJ, doi:10.7717/peerj.19085_

## Round 0.1 · original submission · Major Revisions

Dear Dr. Yu,

Thank you for your submission to PeerJ.

After getting the comments from the reviewers, my suggestion for this article is it requires extensive revision including rephrasing the title of this manuscript.

Reviewer 1 ·

Basic reporting

1. “Chaetoglobosin A induces T-24 apoptosis in human bladder cancer cells through oxidative stress and MAPK/PI3K-AKT-mTOR pathway”. The title of the study is very confusing. Please make it simple like “Chaetoglobosin A induces apoptosis in human T-24 bladder cancer cells through oxidative stress and MAPK/PI3K-AKT-mTOR pathway”.
2. In the section “ChA inhibits T-24 cell migration”, Fig 2A shows that within 24h period control cells completely occupy all the scratch areas. It seems T-24 cell proliferation/migration rate is very fast.
 What is the proliferation rate of T-24 cells? I suggest conducting MTT assay at a different time like 0h, 12h, 24h, 48h, and 96h.
 I suggest repeating this experiment and including measurement at 12h also. So, it will be like 0h, 12h, and 24h measurements.
 Line 187-188, “And the distance of cell migration was significantly increased in the untreated group compared to the drug-treated group”
- Please correct this sentence and make it simple.
3. ChA induces apoptosis in T-24 bladder cancer cells. Is this finding T-24 cell line specific or do other bladder cancer cells also show the same trend?
4. Authors found ChA induces ROS in T-24 bladder cancer cells. Does ChA also induce ROS-mediated apoptosis in normal cells? Please mention this in the discussion with citation. Or authors should conduct MTT assay and ROS generation assay in at least in a normal cell line.
5. Under the section “Effect of ChA on apoptosis-associated proteins in T-24 cells” in Figure 6A, authors should show the expression of cleaved caspase3 and cleaved caspase8 by western blotting after ChA treatment. Please repeat BCL2 western blotting, it is very difficult to see expression changes in the present figure.
6. Please provide the catalog number of all antibodies used in this study. Secondary Anti-mouse and anti-rabbit antibodies were used, what about primary antibodies? Please provide information about primary antibodies in the method.
7. In Abstract lines 35-36, “These suggest that regulation of oxidative stress, MAPK and PI3K-AKT-mTOR signaling pathways affect apoptosis”. This line is meaningless here as it is a general statement. This study is based on a single cell line treated with ChA. This is a concluding line, so please make it correct like, “These suggest that ChA induces apoptosis by regulating oxidative stress, MAPK, and PI3K-AKT-mTOR signaling pathways in T-24 cells”.
8. Figure 3 legend, (BA) should be (B).
9. Line 197-198, for the statement “the apoptotic morphological features of cell crumpling and chromatin condensation”; please provide reference.
10. Line 205-206, please provide reference.
11. Figure 4 legend (A), “ROS production in T-24 cells treated with DCFH-DA for 2 h before staining with ChA”; this sentence is not clearly understood. Please make it simple.
12. Line 210-211, for the statement “The decrease in the MMP is an event that occurs early in apoptosis”. Please provide a reference.
13. In the section, “ChA induces ROS accumulation and MMP reduction in T-24 cells”; please mention which figure the authors refer to. Line 205-210, Fig 4A-B. line 210-214, Fig 4C-D. line 213, this section refers to Fig4, not Fig3; please correct it.
14. Line 213-214, how authors quantified “loss of MMP%”. Please mention that in its respective ‘Method’ section.
15. In the section, “ChA induces T-24 cell cycle blockade”; please mention which figure the authors refer to.
16. Line 226-227, ‘As shown in the Figure 5’ should be ‘As shown in the Figure 5C-D. please correct it for all the result sections.
17. Line 231-232, 243-247, 255-257; please provide references.
18. Figure 7C and D, ratio of p-PI3K/PI3K, p-AKT/AKT is increased, whereas p-mtor/mtor is decreased. Is it consistent with previous findings in bladder cancer? Please explain it with citation in discussion.
19. For the result sections, authors should refer to the figure for each finding. Please make it consistent throughout the results.

Experimental design

No comment

Validity of the findings

No comment

Additional comments

No comments

Reviewer 2 ·

Basic reporting

Jia Song et.al., have evaluated the effect of Chaetoglobosin A in human bladder cancer cells (T-24) and have reported that it induces apoptosis in T-24 cells through oxidative stress and via MAPK/PI3K-AKTmTOR pathway. At the very onset I found the title of the manuscript a little bit confusing. I feel that the title should be rephrased and written in a way that is clear enough to understand. The current title is “Chaetoglobosin A induces T-24 apoptosis in human bladder cancer cells through oxidative stress and MAPK/PI3K-AKTmTOR pathway”, instead if it is written as “Chaetoglobosin A induces apoptosis in human bladder cancer cells through oxidative stress and MAPK/PI3K-AKTmTOR pathway”, it might sound clear enough. The authors have performed extensively for this manuscript, however, there are some major points that needs to be addressed in order to consider this manuscript for publication.

The authors should improve the introduction part and overall writing.

Experimental design

1. Figure 1A & 1C are not aligning with each other. For instance, in Figure 1B cell viability assay by MTT at 12.5 micromolar concentration of ChA, there’s about 85% viability. In contrast, in the colony formation assay, there is complete cell death at the same concentration as no colonies are visible at that concentration.
2. The Figure 1C, colony formation assay should be repeated as the control well itself is not convincing and a representative better image should be provided.
3. Also, the number of colonies should be quantified and a histogram should be plotted.
4. In the manuscript, all the figures illustrating the results are too small to read. It would be fantastic if the authors could increase the resolution of the images so that it can be distinctly understood.
5. The supplementary data does not contain figure legends. It should be mentioned properly with proper labeling of each data.
6. The image quality & resolution of all the figure panels should be improved.

Validity of the findings

The authors have performed all the experiments in only one cell line and no in vivo studies are done. Therefore, if it could be repeated in a few more cell lines than the impact of the anti-tumor compound would be more scientifically evident.

Additional comments

The authors should thoroughly revise the sentence formation and grammatical errors throughout the manuscript. There are numerous problems in the sentence formation which did not deliver the proper meaning of the author’s point of view.

Annotated reviews are not available for download in order to protect the identity of reviewers who chose to remain anonymous.

---

## Round 0.2 · accepted · Accept

Thank you for your submission to PeerJ.

Your revised manuscript - Chaetoglobosin A induces apoptosis in T-24 human bladder cancer cells through oxidative stress and MAPK/PI3K-AKT-mTOR pathway - addressed the reviewer's questions and has been Accepted for publication. Congratulations!

Reviewer 2 ·

Basic reporting

No comment

Experimental design

No comment

Validity of the findings

No comment